# Completing the loop of the Late Jurassic–Early Cretaceous true polar wander event

Yifei Hou[1], Pan Zhao[1] ✉, Huafeng Qin[1], Ross N. Mitchell[1], Qiuli Li[1], Wenxing Hao[1], Min Zhang[2], Peter D. Ward[3], Jie Yuan[1], Chenglong Deng[1] & Rixiang Zhu[1]

The reorientation of Earth through rotation of its solid shell relative to its spin axis is known as True polar wander (TPW). It is well-documented at present, but the occurrence of TPW in the geologic past remains controversial. This is especially so for Late Jurassic TPW, where the veracity and dynamics of a particularly large shift remain debated. Here, we report three palaeomagnetic poles at 153, 147, and 141 million years (Myr) ago from the North China craton that document an ~12° southward shift in palaeolatitude from 155–147 Myr ago (~1.5° Myr⁻¹), immediately followed by an ~10° northward displacement between 147–141 Myr ago (~1.6° Myr⁻¹). Our data support a large round-trip TPW oscillation in the past 200 Myr and we suggest that the shifting back-and-forth of the continents may contribute to the biota evolution in East Asia and the global Jurassic–Cretaceous extinction and endemism.

True polar wander (TPW) is the reorientation of the entire solid crust-mantle shell of a planet with respect to its rotation axis[1]. It arises from centrifugal forces acting on mass anomalies either on the surface or within the body of a quasi-rigid planet, which on Earth is a long-term process with the solid Earth shifting in a secular manner beneath its spin axis[2]. TPW currently happening today, documented with astronomical observations for over a century and with satellites for several decades, occurs at a rate of ~1° Myr⁻¹ and is thought to be caused by a combination of Holocene deglaciation and longer timescale mantle processes[3–6]. Comparison of successive high-quality palaeomagnetic poles is an effective means of testing TPW and multiple episodes of large-amplitude TPW spanning the Palaeoproterozoic to the Cretaceous have been revealed[7–12]. However, the occurrence of TPW in the geologic past remains highly controversial[13–15]. It is during the past 300 Myr since supercontinent Pangaea and its breakup when palaeogeography is most accurately known that TPW events are most keenly testable.

The Mesozoic Era appears to have potentially been an active time interval for TPW, but large uncertainties remain. According to global palaeomagnetic analyses, a ~18° counterclockwise TPW rotation of supercontinent Pangaea occurred from 250–220 million years ago (Ma) around an equatorial axis located in (modern) western Africa (0°N, 11°E; ref. 12), followed by a clockwise rotation of the same amount between 195 and 145 Ma (refs. 12,16,17). The latter event, the clockwise Jurassic TPW rotation during the beginning of Pangaea breakup, has since been revealed by several palaeomagnetic studies nearly globally, however, both the veracity and speed of this TPW event is hotly debated[8,16–25]. One viewpoint argues for a phase of rapid rotation between 160 and 145 Ma at rates of 1.5–2.5° Myr⁻¹, which is called the Late Jurassic "monster shift"[8,18,19,21–25] (Fig. 1). Another viewpoint questions much of the palaeomagnetic data underpinning the existence of the "monster shift" and instead asserts a steady continental rotation at a rate of ≤0.8° Myr⁻¹ throughout the Jurassic[17,20] (Fig. 1). Differences in the maximum rate and duration of TPW imply dramatically different geodynamical conditions in the mantle and the lithosphere and/or different shapes of Earth's nonequilibrium figure[7].

In order to address these competing models, it is therefore critical to acquire new well-dated, high-resolution and high-quality

[1]State Key Laboratory of Lithospheric Evolution, Institute of Geology and Geophysics, Chinese Academy of Sciences, Beijing 100029, China. [2]Key Laboratory of Earth and Planetary Physics, Institute of Geology and Geophysics, Chinese Academy of Sciences, Beijing 100029, China. [3]Department of Biology, University of Washington, Seattle, WA 98995, USA. ✉e-mail: panzhao@mail.iggcas.ac.cn

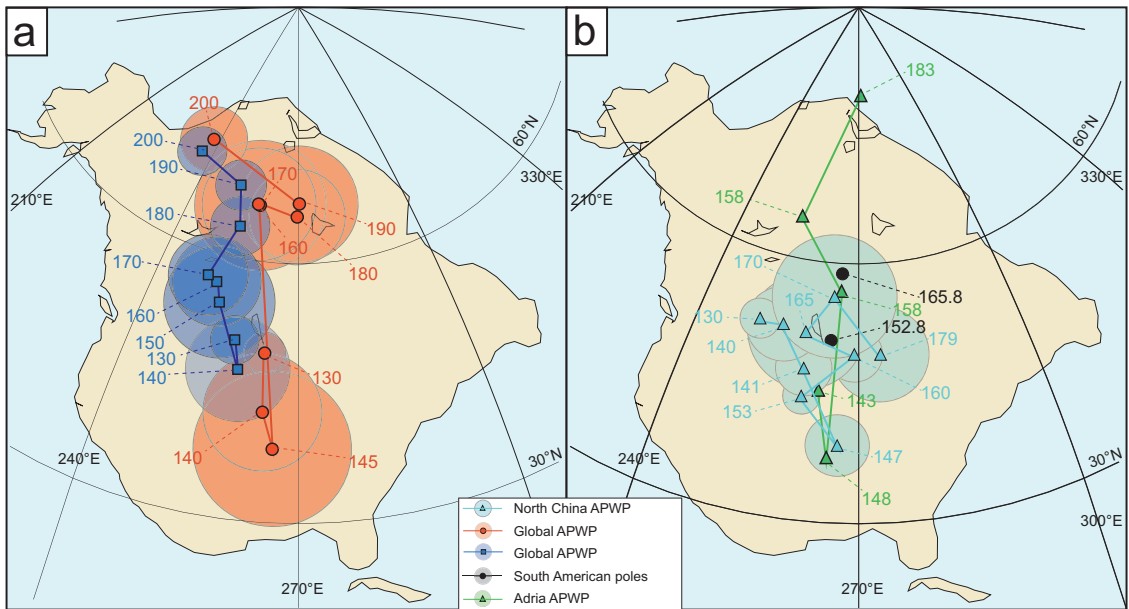

**Fig. 1 | Apparent polar wander paths (APWPs) showing possible Late Jurassic TPW. a** Contrasting global APWPs from Torsvik et al.[16] and Kent and Irving[19] in northwest African coordinates. **b** The APWPs from Adria[25], South America[18] and North China (this study) in northwest African coordinates. Poles used are presented in Supplementary Table 1. Due to the North China cratons were excluded from supercontinent Pangaea during Jurassic period, we manually rotated the APWP of the North China craton to northwest African coordinates using rotation parameter (353.5, 1.2, 32.9). The APWPs compiled by Kent and Irving[19] and the Muttoni and Kent[25] use Euler rotation parameters from Kent and Irving[19]. The APWPs compiled by Fu et al.[18] and Torsvik et al.[16] use Euler rotation parameters from Torsvik et al.[16].

palaeomagnetic data during the critical 160−140 Ma interval in question. As a global process, TPW must be recorded by all plates. During the Jurassic, the North and South China cratons were excluded from supercontinent Pangaea[26]. Thus, East Asia therefore provides an additional palaeogeographic vantage from which to test TPW. The possibility of Late Jurassic TPW has not been adequately studied in the tectonic region, which can be ascribed to the lack of reliable Late Jurassic (160−145 Ma) palaeomagnetic poles from the various cratons of East Asia.

In our study of the North China craton (NCC), we present palaeomagnetic results from the continuous volcano- and clastic-stratigraphic sequence of the Late Jurassic Tiaojishan Formation and the Late Jurassic−Early Cretaceous Tuchengzi Formation (Fig. 2; Supplementary Figs. 1 and 2). Three robust and well-dated palaeomagnetic poles at critical ages of 153, 147, and 141 Ma were obtained from the NCC (Supplementary Table 3). Our results demonstrate the existence of the Late Jurassic "monster shift" in East Asia, definitively establishing the rapidity of the TPW event by documenting fast rates of continental motion, as well as arguing for an underappreciated subsequent return trip that yields a complete round-trip TPW oscillation. We consider the implications of the identified large-scale TPW oscillation from its possible geodynamic and tectonic forcing to its potential promotion of the Jurassic−Cretaceous biotic evolution and extinction.

## Results
### Zircon U−Pb geochronology
Zircon U−Pb geochronological constraints were acquired in this study from a continuous volcano- and clastic-stratigraphic section studied for palaeomagnetism (Fig. 2a). Standard zircon geochronological U−Pb SIMS and LA-ICP-MS laboratory and analytical methods were used and are described in detail in the Methods. For volcanic rock sample TJS-TC1 from the top of the Tiaojishan Formation, a weighted mean $^{206}Pb/^{238}U$ age of $152.9 ± 2.5$ Ma (SIMS, $2\sigma$, $n = 8$, MSWD = 0.3; Fig. 2a and d; Supplementary Table 2 and 3) was obtained,

representing the cooling age of the volcanic rocks sampled for palaeomagnetism at this lower stratigraphic level of the section. Two samples from the top of the Tuchengzi Formation collected from a pyroclastic rock yielded consistent $^{206}Pb/^{238}U$ weighted mean ages of $141.8 ± 1.1$ Ma (SIMS, $2\sigma$, $n = 28$, MSWD = 0.9) for sample TCZ-TA1 and $141.1 ± 1.7$ Ma (SIMS, $2\sigma$, $n = 23$, MSWD = 1.6) for sample TCZ-TB1 (Fig. 2b and c; Supplementary Tables 2 and 3). For sample TCZ-M1 from red sandstone from the middle part of the Tuchengzi Formation, two youngest detrital zircon grains give an age of 151 Ma (LA-ICP-MS) providing a maximum depositional age (Fig. 2e). Another pyroclastic sample (TCZ-M2) collected below sample TCZ-M1, which should be younger than 151 Ma, yield continuous zircon ages (SIMS) from $166.4 ± 4.7$ to $147.1 ± 3.9$ Ma (Fig. 2f; Supplementary Tables 2 and 3), indicating abundant inherited zircons. Therefore, we use the youngest three ages to calculate a weighted mean $^{206}Pb/^{238}U$ age of $147.5 ± 4.5$ Ma ($2\sigma$, $n = 3$, MSWD = 0.03; Fig. 2g), which represents the cooling age of this pyroclastic layer. Thus, we conclude that the best estimate of the TCZ-M section should be ca. 147 Ma.

### Palaeomagnetism
In total, 463 palaeomagnetic samples were collected from the continuous Late Jurassic−Early Cretaceous volcano- and clastic-stratigraphic sequence (from bottom to top): 195 volcanic and pyroclastic samples from the upper Tiaojishan Formation, 120 red sandstone samples from the middle of the Tuchengzi Formation, and 148 pyroclastic and tuffaceous sandstone samples from the top Tuchengzi Formation. Standard palaeomagnetic field, laboratory, and analytical methods were used and are described in detail in the Methods. Rock magnetic investigations reveal that magnetite is the main magnetic remanence carrier for the volcanic and volcanoclastic samples from the upper Tiaojishan Formation and the upper Tuchengzi Formation, whereas hematite is the main magnetic carrier for the red sandstone samples from the middle part of the Tuchengzi Formation (Supplementary Fig 4).

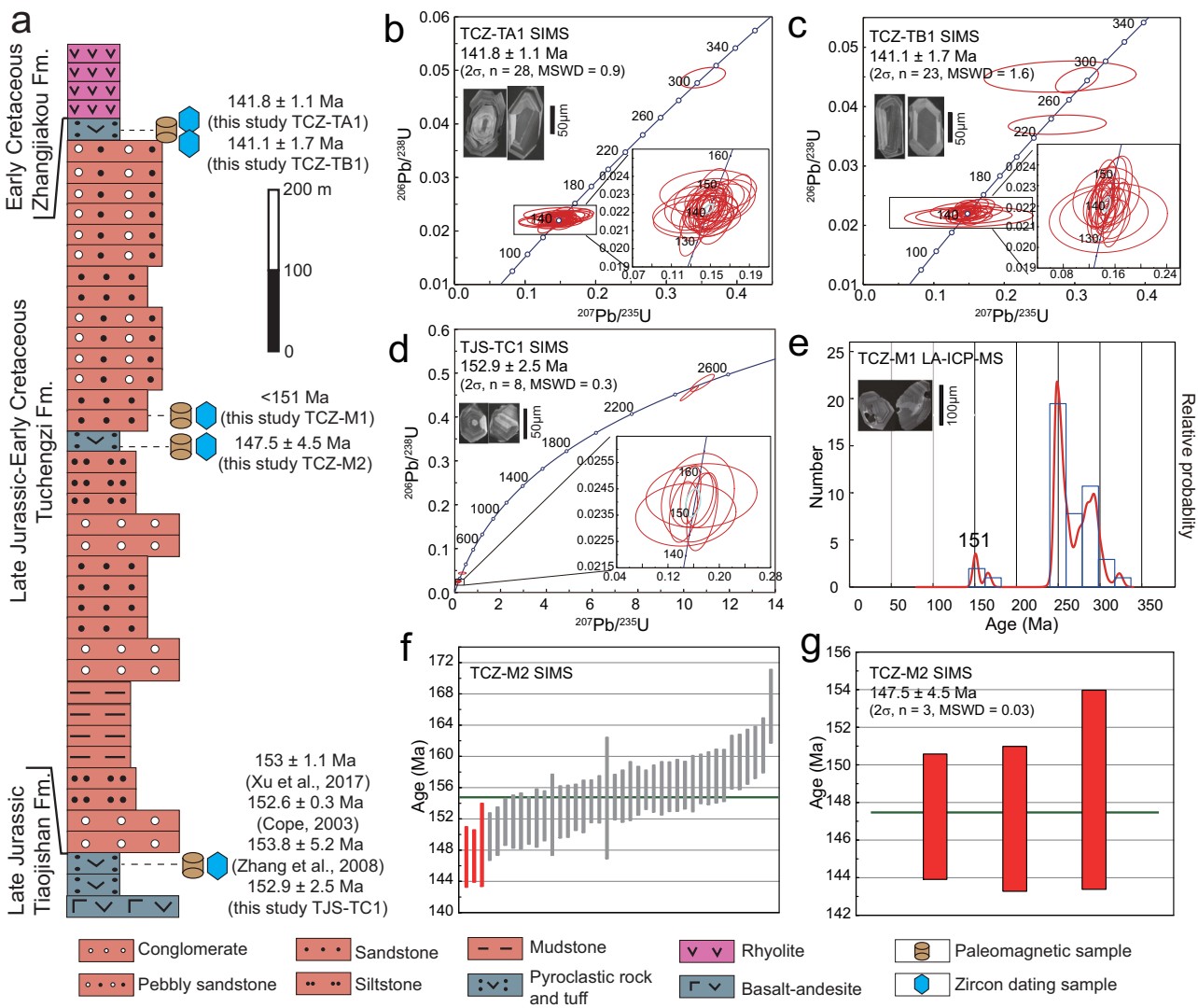

**Fig. 2 | Lithostratigraphy and zircon U–Pb chronology. a** The Late Jurassic−Early Cretaceous sequence containing three formations with zircon U–Pb ages indicated. **b**–**d** SIMS U–Pb zircon dating results of samples from the top of the Tuchengzi Formation and the top of Tiaojishan Formation. **e** Detrital zircon LA-ICP-MS U–Pb geochronologic results for a sample collected from the middle part of the Tuchengzi Formation. **f** SIMS U–Pb zircon dating results of samples from the middle of the Tuchengzi Formation and (**g**) its primary zircon results.

## Upper part of the Tiaojishan Formation (153 Ma)

Specimens from the three sections (TJS-TA, TJS-TB, and TJS-TC) spanning the upper Tiaojishan Formation exhibit similar demagnetization behavior characterized mainly by two magnetic components (Supplementary Figs. 5a–c). The low-temperature components (LTC), isolated in temperature ranges up to ~200 °C or ~400 °C, yielded a mean direction (in geographic coordinates) with declination ($D_g$) = 353.2°, inclination ($I_g$) = 59.0°, and a cone of 95% confidence ($\alpha_{95}$) = 2.3°, which is statistically indistinguishable from that of the present-local field in the sampling location and can be considered as a present-day overprint due to weathering (potentially carried by goethite up to ~150 °C) and/or a viscous remanent magnetization (VRM) carried by pseudo-single-domain and/or multi-domain magnetite. After removal of the LTC, the high-temperature components (HTC) display normal and reversed polarities that can be considered as the characteristic remanent magnetization (ChRM). The high-temperature components of section TJS-TA displayed a mean direction with $D_g = 112.8°$, $I_g = 43.9°$ ($k_g = 37.3$, $\alpha_{95g} = 2.4°$) before and $D_s = 36.8°$, $I_s = 50.6°$ ($k_s = 69.7$, $\alpha_{95s} = 1.7°$) after tilt correction (Supplementary Fig. 6a). The ChRMs of section TJS-TB was calculated at $D_g = 49.2°$, $I_g = 70°$ ($k_g = 50.6$, $\alpha_{95g} = 3.3°$) before and $D_s = 6.4°$, $I_s = 45.4°$ ($k_s = 41.7$, $\alpha_{95s} = 3.7°$) after tilt correction (Supplementary Fig. 6b). For TJS-TC section, the ChRMs of specimens show normal and reversed polarities but not antipodal, which might result from limited number of specimens and effect of palaeosecular variation. Nevertheless, a mean direction was calculated at $D_g = 44.3°$, $I_g = 49.5°$ ($k_g = 22.2$, $\alpha_{95g} = 5.7°$) in geographic and $D_s = 17.4°$, $I_s = 56.1°$ ($k_s = 22.2$, $\alpha_{95s} = 5.7°$) in stratigraphic coordinates (Supplementary Fig. 6c).

Note that the inclinations from the three sections are similar, but there is slight difference for the declinations of section TJS-TA from sections TJS-TB and TJS-TC. The mean direction of sections TJS-TB and TJS-TC is $D_s = 10.7°$, $I_s = 50.3°$ ($k_s = 25.7$, $\alpha_{95s} = 3.5°$) in stratigraphic coordinates (Supplementary Fig. 6d). The difference in declination between the mean direction and declination of section TJS-TA is 26.1°. By plotting declinations of Jurassic poles from the NCC, we found that declinations are consistent, including the declinations of sections TJS-TB and TJS-TC from this study (Supplementary Fig. 7), indicating that the NCC as a rigid craton experienced no obvious self-rotation during the Jurassic. However, the declination of section TJS-TA is significantly different from others (Supplementary

Fig. 7), which is reasonable to be ascribed to local vertical-axis rotation related to local strike-slip faulting and block rotation (Supplementary Fig. 2c). Meanwhile, such a magnitude and clockwise sense of rotation is consistent with the bedding strike directions of the sections, with the TJS-TB and TJS-TC sections N-striking and the TJS-TA section NE-striking (Supplementary Fig. 2c, d, and f). Comparing with the 160 Ma poles of the NCC, the mean declination calculated from sections TJS-TB and TJS-TC show a declination change of $9° \pm 4°$, which is in the same range as predicted declination changes with APWPs from both Kent at al.[8] ($17° \pm 4°$) and Torsvik et al.[16] ($6° \pm 4°$) when errors are considered (Supplementary Fig. 7). On the contrary, the data from section TJS-TA show a much larger declination change ($35° \pm 3°$), which is inconsistent with the declination change in the TPW framework (Supplementary Fig. 7). Therefore, we ascribe the declination change of section TJS-TA to local vertical axis rotation. We thus rotated the magnetic directions of the TJS-TA section 26.1° counterclockwise and calculated a combined age-mean direction for the upper Tiaojishan Formation (ca. 153 Ma) at $D_g = 93.5°$, $I_g = 66.0°$ ($k_g = 12.2$, $\alpha_{95g} = 3.3°$) in geographic, and $D_s = 10.7°$, $I_s = 50.5°$ ($k_s = 41.1$, $\alpha_{95s} = 1.7°$) in stratigraphic coordinates (Fig. 3a). The concentration parameter $k$ of this mean direction is significantly increased after tilt correction, which passes a McElhinny[27] fold test at the 99% confidence level. The stepwise unfolding approach of Watson and Enkin[28] reveals a $k_{max}$ value at 91.5% unfolding, indicating a pre-folding acquisition of magnetic remanence (Fig. 3d). Therefore, the ChRMs of the upper Tiaojishan Formation (153 Ma) are primary, and a ca. 153 Ma palaeomagnetic pole was calculated for the NCC at 77.3°N, 249.1°E ($A_{95} = 2.1°$, $n = 162$) (Supplementary Table 5).

## Middle part of the Tuchengzi Formation (ca. 147 Ma)

Two components were isolated for most specimens (Supplementary Fig. 5d). LTCs with directions close to the present-local field direction are interpreted as recent overprints, where a chemical remanent magnetization (CRM) is most likely required to explain their persistence to temperatures as high as ~540 °C (exceeding the unblocking temperature of goethite and typical VRM stability). After removing the LTC, a majority of samples show a linear decay of the remanence to the origin until 685 °C, indicating this ChRM direction is a post-depositional remanent magnetization (pDRM) carried by hematite, consistent with our rock magnetic experiments. ChRM directions were obtained from 103 out of 120 red sandstone specimens, with 92 specimens of normal polarity and the other 11 specimens of reversed polarity (Supplementary Table 4). ChRMs of 10 specimens of pyroclastic rock show dispersed directions that are also inconsistent with those from red sandstone specimens (Supplementary Table 4) and they were excluded from further calculations. Combining the 103 ChRM directions, a mean direction was calculated at $D_g = 20°$, $I_g = 17°$ ($k_g = 15.6$, $\alpha_{95g} = 3.6°$) before, and at $D_s = 9.3°$, $I_s = 38.4°$ ($k_s = 15.6$, $\alpha_{95s} = 3.6°$) after tilt correction (Fig. 3b).

As the samples come from monoclinal strata, a fold test cannot be performed. However, the ChRM directions reveal antipodal normal and reversed polarities that pass C-class reversal test[29], arguing for a primary magnetization for the middle part of the Tuchengzi Formation. In such red sandstone lithologies, inclination shallowing should be tested and corrected for before using their ChRM direction for tectonic interpretation. Using the elongation/inclination (E/I) correction that is based on the PSV model TK03 (refs. [30–32]), a flattening factor $f = 0.85$ (Supplementary Fig. 8; $\tan I_c = f * \tan I_o$, where $I_c$ and $I_o$

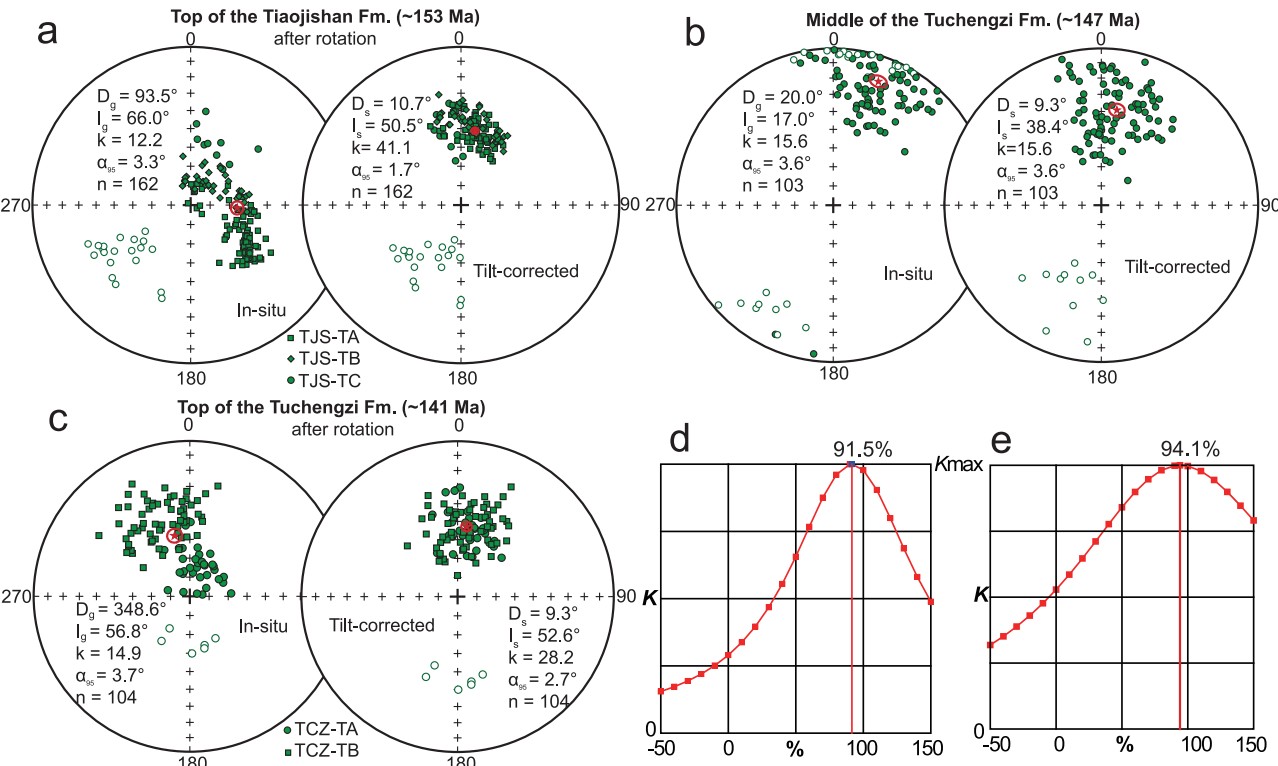

**Fig. 3 | Palaeomagnetic results.** Equal-area projections of the palaeomagnetic directions of the high-temperature components (green), showing mean directions from each section (red). Statistical results, stratigraphically from bottom to top, for (**a**) combined specimens from three sections (TJS-TA, TJS-TB, TJS-TC) of the top of the Tiaojishan Formation (ca. 153 Ma) after rotating directions of section TJS-TA - 26.1° counterclockwise relative to the union of sections TJS-TB and TJS-TC, (**b**) the middle part of the Tuchengzi Formation (ca. 147 Ma), (**c**) combined specimens from two sections (TCZ-TA and TCZ-TB) of the top of the Tuchengzi Formation (ca. 141 Ma) after rotating directions of section TCZ-TB - 43° counterclockwise relative to section TCZ-TA. Results were calculated with 95% confidence limits. Progressive unfolding[28] of specimen-mean directions showing a maximum directional clustering ($K_{max}$) at 91.5% untilting for the top of Tiaojishan Formation after rotation (**d**) and at 94.1% untilting for the top of Tuchengzi Formation after rotation **e**.

represent the measured and original inclinations) was obtained, which is consistent with the $f$ value ($f = 0.9$) obtained from the red sandstone of the Tuchengzi Formation by Ren et al.[33]. With this flattening factor, the mean inclination was corrected from 38.4° to 42.6° with 95% confidence limits between 37.6° and 48.6° and a ca. 147 Ma palaeomagnetic pole was calculated for the NCC at 72.3° N, 268.2° E ($A_{95} = 3.6°$, $n = 103$) (Supplementary Table 5).

## Top of the Tuchengzi Formation (141 Ma)

For the TCZ-TA section in the upper Tuchengzi Formation, most specimens exhibit a single magnetic component typically stable up to temperatures of ~620 °C (Supplementary Fig. 5e). Antipodal normal ($n = 35$) and reversed ($n = 6$) polarity ChRM directions were identified and combined to yield a mean ChRM direction (at $D_g = 10.7°$, $I_g = 72.5°$) ($k_g = 39.2$, $\alpha_{95g} = 3.6°$) in geographic, and at $D_s = 6.8°$, $I_s = 52.6°$ ($k_s = 39.2$, $\alpha_{95s} = 3.6°$) in stratigraphic coordinates (Supplementary Fig. 6e). Again a fold test cannot be conducted as the samples derive from a monocline, but a positive B-class reversal test[29] argues for the primary nature of the magnetization. For the TCZ-TB section of the upper Tuchengzi Formation, one-third of the measured specimens exhibit two components. After removing the LTCs, the HTCs, mostly calculated from 400–620 °C, show uniformly normal polarity ChRM directions (Supplementary Fig. 5f). In total, 63 specimens yield a mean direction at $D_g = 16.3°$, $I_g = 63°$ ($k_g = 23.7$, $\alpha_{95g} = 3.8°$) in geographic, and at $D_s = 54°$, $I_s = 52.5°$ ($k_s = 23.7$, $\alpha_{95s} = 3.8°$) in stratigraphic coordinates (Supplementary Fig. 6f).

The inclinations of the mean directions in stratigraphic coordinates of the two sections are consistent, but there is a ~ 47° difference in declination, which likely results from the local structural vertical-axis rotation of one of the two sections. Anisotropy of magnetic susceptibility (AMS) results (Supplementary Fig. 9) suggest that the TCZ-TB section experienced a ~ 43° clockwise rotation with respect to the TCZ-TA section, which is indistinguishable within uncertainty from the difference in declination. With only one set of transcurrent faults (as observed in the field area; Supplementary Fig. 2e), such an amount of block rotation is within the theoretically permissible upper limit[34]. Therefore, we rotated the magnetic directions of the TCZ-TB section 43° counterclockwise and calculated a combined age-mean direction for the ChRM directions from both sections of the ca. 141 Ma upper Tuchengzi Formation (in geographic coordinates) at $D_g = 346.0°$, $I_g = 57.0°$ ($k_g = 14.6$, $\alpha_{95g} = 3.8°$) before tilt correction, and (in stratigraphic coordinates, i.e. after tilt correction) at $D_s = 7.3°$, $I_s = 52.5°$ ($k_s = 28.6$, $\alpha_{95s} = 2.6°$) (Fig. 3c). Multiple fold test algorithms[27,28] yield positive tests for the combined mean direction of the two sections with the $k_{max}$ value at 94.1% unfolding, arguing for the magnetization being pre-folding in age and therefore most likely primary in origin (Fig. 3e). Furthermore, this mean direction from our study is close to that obtained from the upper Tuchengzi Formation from the Beipiao Basin[33], supporting its primary nature and the structural correction for one of our two sections. Therefore, a reliable ca. 141 Ma palaeomagnetic pole was calculated for the NCC at 80.4°N, 244.1°E ($A_{95} = 3.2°$, $n = 104$) (Supplementary Table 5).

## Discussion

The indication of Jurassic TPW has long been a feature of global apparent polar wander paths (APWPs) ever since they were constructed using the well-known relative Euler rotations of the major continents since 200 Ma (refs. 35,36). Early APWPs suggested a total of ~30° of TPW for the last 200 Myr with periods of (quasi) standstill alternating with faster TPW[36]. Steinberger and Torsvik[12] used a different approach of visualizing wholesale rotations of the continents and proposed a steady clockwise TPW rotation from 195 to 145 Ma around a Euler pole near the center of the African large low shear-wave velocity province (LLSVP) and the antipodal Pacific LLSVP, where these largest mass anomalies on the planet tend to control the TPW axis defined by

Earth's equatorial minimum moment of inertia[37]. However, updated APWPs led to a revised model of this TPW event as having a steady phase between 195 and 150 Ma and an accelerated phase from 150 to 140 Ma (ref. 16). With a focus on careful assessment of poles from North America, the best-sampled APWP globally, Kent and Irving[19] also constructed revised global APWP and argued for a standstill period from 190 to 160 Ma followed by a fast shift of ~30° between 160 and 145 Ma, which they termed the Late Jurassic "monster shift".

Since its identification, the putative "monster shift" has been suggested to be further supported by Late Jurassic palaeomagnetic poles reported from North America, South America, Adria, and the Pacific plate[8,18,21,25]. These recent detailed studies suggest a fast rate of motion of 1.5–2.5° Myr$^{-1}$ for both continental and oceanic plates, strongly suggestive of TPW. Based on comparison of the observed rotation rate with the expected rates of TPW and relative lithosphere–mantle motion, Fu et al.[15] argued that the "monster shift" was an episode of TPW, i.e., the shift resulted from excitation from mantle convection and the net rotation of the lithosphere relative to the mantle is negligible. Most recently, a Late Jurassic palaeomagnetic pole obtained from dikes from Greenland with a mean age of 147.6 ± 3.4 Ma was reported, which is interpreted to not support the Late Jurassic "monster shift" but instead suggested to indicate a steady polar motion with rates of ~0.7° Myr$^{-1}$ (ref. 20), consistent with that proposed by Torsvik et al.[16]. Furthermore, it is argued that in order to not violate younger poles after the supposed shift, the "monster shift", if valid, should be followed by a counterclockwise return-trip TPW oscillation between 147 Ma and 138 Ma (ref. 20), calling into question the event. In fact, Muttoni and Kent[25] did identify a slower retrograde polar motion of about 10° in ~10 Myr occurring from 148 to 143 Ma following the "monster shift" (Fig. 1), which continued until the start of the Cretaceous standstill; however, it is ambiguous whether this retromotion is the result of plate motion of North America or a return-leg episode of TPW. Recently, by studying the palaeomagnetism of the Lhasa terrane, Ma et al.[38] suggested a yoyo-like drift motion that supports the Late Jurassic "monster shift". However, Li et al.[39] suggested palaeolatitudinal standstill of Lhasa terrane, due to northward plate movement and southward TPW were at similar velocity of ~0.8° Myr$^{-1}$ in the Jurassic. Thus, they argue for steady TPW supporting the opinion of Torsvik et al.[16]. Furthermore, a new global APWP calculated from site-level data also did not recognize the "monster shift" either[40]. The discrepancy between our study and Vaes et al.[40] may be attributed to the calculation window. In this current study, the APWP calculated in a 10 Myr window shows a higher APW rate (~0.6°/Myr) than when calculated in a 20 Myr window (~0.2°/Myr) around 150 Ma[40]. Further, the Late Jurassic TPW may not have been identified owing to the omission of three key poles (the 169 Ma Moat pole, the 155 Ma Peddie pole and the 147 Ma Ithaca pole). In addition, while TPW is a global event, its local record can be easily influenced by the additional tectonic movement of an independent plate. Therefore, although our work supports a Late Jurassic "monster shift", it remains controversial and requires testing from more individual plates globally.

Recent palaeomagnetic studies of East Asia have attempted to conduct such a definitive test of the monster shift[41,42]. Based on two palaeomagnetic poles recently obtained from the NCC, a large and fast TPW event has been proposed from 174 to 157 Ma, which caused a rapid ~25° southward movement of East Asian blocks[42]. However, the age span of this supposed fast TPW event is largely distinct from and predates the 160−145 Ma "monster shift"[8]. Furthermore, three palaeomagnetic poles were subsequently obtained from volcanic layers from the northern margin of the NCC dated at 170 Ma, 165 Ma, and 160 Ma that argued that East Asia did not undergo any significant southward shift between 170 and 160 Ma (ref. 41). Thus, to date, tests of the "monster shift" in East Asia have been ambiguous.

Combining our three palaeomagnetic poles at 153 Ma, 147 Ma, and 141 Ma with strictly selected high-quality Jurassic−Cretaceous

palaeomagnetic poles of Gao et al.[41], we constructed a Jurassic–Cretaceous APWP for the NCC (Supplementary Table 5) and calculated the corresponding palaeolatitude of the NCC (reference point of 41° N, 121° E; Fig. 4a, b). The overlapped poles and consistent palaeolatitude in the age of 180−160 Ma indicate a standstill stage of East Asia (Supplementary Table 5; Fig. 4a, b), which can be interpreted as the counteraction of northward plate motion due to subduction of the Mongol–Okhotsk Ocean and southward motion result from the TPW[36].

After the 180-160 Ma standstill of East Asia, a significant southward displacement of ~12° from 155-147 Ma has been revealed with an average latitudinal velocity of 1.53° Myr⁻¹ (Fig. 4a). This southward shift is comparable with that revealed by the APWP constructed by Kent et al.[8] (Fig. 4a, b) so we argue that this southward shift represents an episode of TPW. Then, a northward movement of the NCC in the magnitude of ~15° is recorded between 147 Ma and 141 Ma with a mean velocity of 1.61° Myr⁻¹, which is consistent with the northward motion

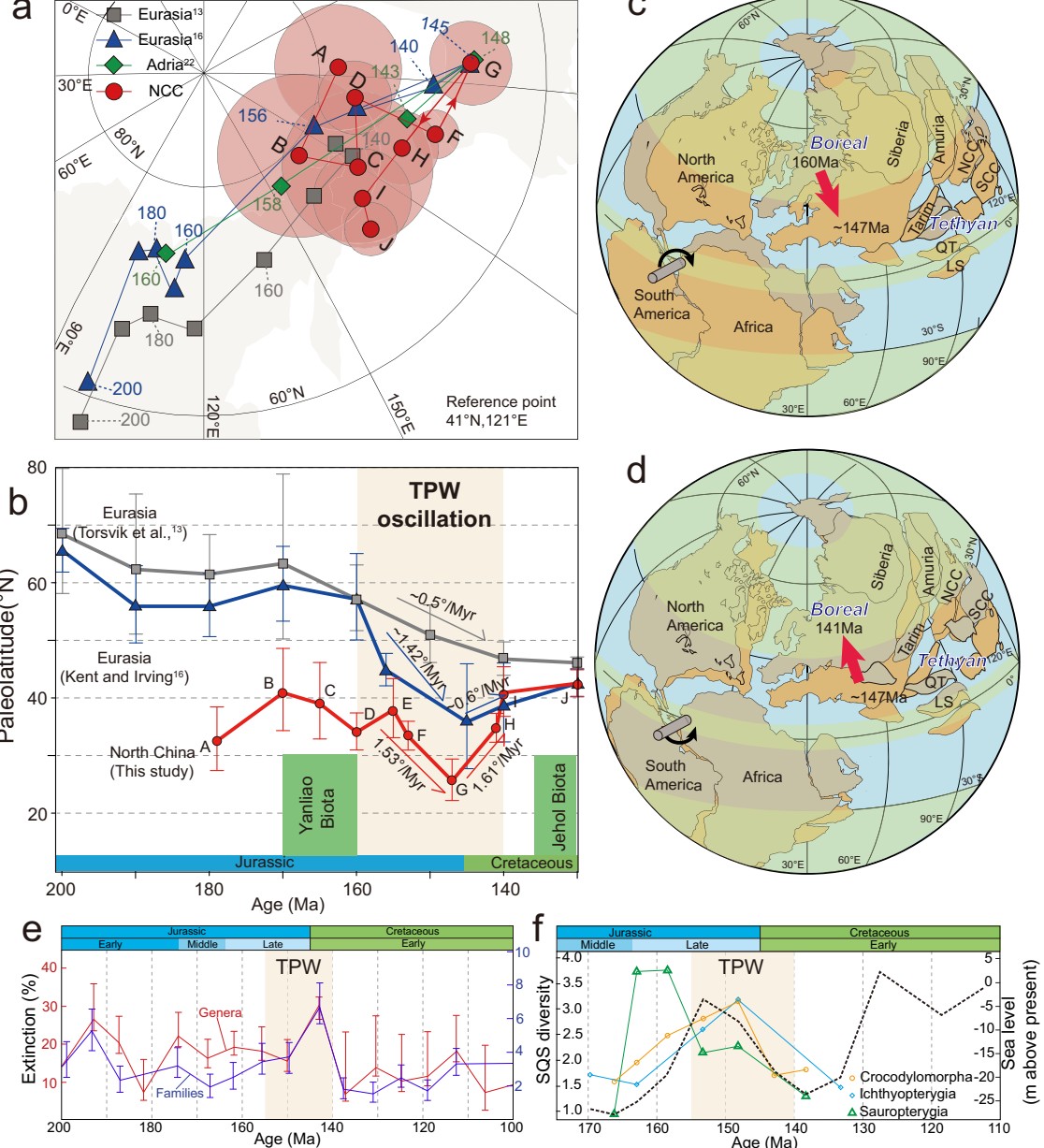

**Fig. 4 | Late Jurassic–Early Cretaceous TPW round-trip oscillation and biogeographic implications.** Mesozoic palaeopoles (in Eurasia coordinates) (**a**) and palaeolatitudinal variation (**b**) for the NCC including those two generated in this study. To better compare the different plate drift process, the APWP of the North China craton was rotated using rotation parameter (93.1°, 71.4°, −31.6°). Labeled area indicate the Late Jurassic-Early Cretaceous TPW round-trip oscillation. Poles are listed in Supplementary Table 5. Note the TPW oscillation across the period boundary, best expressed in our data from NCC. Durations of the Yanliao Biota and Jehol Biota have been marked that occurred before and after the TPW. **c** First segment of the TPW oscillation–the excursion, known as the "monster shift"–from ca. 155–147 Ma. **d** The immediately following second segment of the TPW

oscillation–the return leg–from ca. 147–141 Ma. The equatorial Euler pivot point for both shifts of opposite sense is located in western Africa following Steinberger and Torsvik[12]. Plate reconstructions were made using the plate circuits from Besse and Courtillot[36] and Kent et al.[8]. SCC–South China craton; QT–Qiangtang; LS–Lhasa. Endemic Boreal and Tethyan biogeographic provinces are indicated[53]. The green belts represent humid zonal climate belts and white belts represent arid climate[25]. Plates moving across zonal climate belts affect environment and the living ecosystems of different species. **e** The variational extinction of families and genera from 200–100 Ma, and there is an increase during TPW. **f** Tetrapod diversity and sea level across the Jurassic-Cretaceous boundary[54]. SQS shareholder quorum subsampling.

of Eurasia, arguing itself as a TPW event rather than plate motion due to relative convergence of NCC and Siberia (Fig. 4c, d). This partial retromotion was noted by Muttoni and Kent[25], but they were not sure whether it was merely North America plate motion or TPW. Combining our own and published data from NCC, this retromotion should represent a recovery episode of TPW after the "monster shift" excursion, which would support the drift model of two stages[38]. Therefore, a TPW "round trip" oscillation event occurred in the Late Jurassic–Early Cretaceous with the previous recognized "monster shift" as the first half (Fig. 4c, d). As TPW is driven by imposed mantle mass anomalies that subsequently relax, and/or due to lithospheric elasticity, a TPW excursion is typically modeled as being followed by a return-leg recovery, together comprising a full round-trip TPW oscillation[7], as has also been observed empirically for other TPW events[10,43–45].

Thus, our Late Jurassic palaeomagnetic poles from East Asia demonstrate a Late Jurassic–Early Cretaceous round-trip TPW oscillation of which the "monster shift" constitutes the first leg, the excursion, of the full oscillation. However, more palaeomagnetic studies are still needed to further confirm the global nature of the return-leg phase, the recovery, of the postulated TPW oscillation as the second event is not yet as convincingly demonstrated as is the "monster shift" excursion phase.

Provided these constraints on the kinematics of not only the Late Jurassic "monster shift", but also the Early Cretaceous return trip, we consider the potential geodynamic drivers of both phenomena at these two times. The largest sign change in Earth's degree-2 geoid kernel occurs as a slab sinks from the upper mantle into the lower mantle crossing the mantle transition zone, causing equatorward and poleward TPW (from the respective of the slab location), respectively[45,46]. The most notable slab dynamics during this late stage of supercontinent tenure of Pangaea occurred in East Asia, with the closure of the Mongol–Okhotsk Ocean finally linking East Asia to Pangaea[47]. Seismic tomography identifies a sunken slab sitting on the core–mantle boundary located today precisely where the active margin was at the time of the "monster shift"[48]. As the closure of this ocean occurred across the critically important Jurassic-Cretaceous boundary–the age of the reversal of polar motion–it is conceivable that the shallow slab during the Late Jurassic driving East Asia equatorward during the "monster shift" reversed direction in the Early Cretaceous, as the slab crossed the mantle transition zone, entering the lower mantle, and driving East Asia poleward during the return-leg oscillation.

An alternative, or complementary, mechanism for the recovery phase of the oscillation, lithospheric elasticity[7], is keenly viable for this age as supercontinent Pangaea was still largely intact and had only begun to breakup (Fig. 4). Whereas for younger TPW events later in Pangaea breakup[10] lithospheric elasticity becomes less viable, during the Early Cretaceous return leg following the "monster shift", lithospheric elasticity would have been high with a unified supercontinent and therefore retained a strong "memory" for the previous hydrostatic bulge before the "monster shift", thus possibly causing a subsequent snapback returning to the initial pole position before the excursion.

The Late Jurassic–Early Cretaceous TPW oscillation caused large latitudinal shifts of East Asian continents (Fig. 4c, d), which could help account for the sudden changes of palaeoenvironment and palaeontological evolution that are known from the region. Also globally, this TPW oscillation may have contributed to global palaeoenvironmental change and the Jurassic–Cretaceous extinction. Before the "monster shift", the long-term (180–160 Ma) standstill of East Asia at palaeolatitudes of 35–40°N led to a warm and humid environment, favorable for plants and animals, which might help promote the 168–159 Ma flourishing of the Yanliao Biota[49]. The "monster shift" since 160 Ma caused a ~12° southward shift of East Asia to hot and dry "doldrum" latitudes, and this timing is precisely consistent with the extinction of the Yanliao Biota (Fig. 4b). Therefore, the "monster shift" caused

sudden environmental change might be responsible for this extinction[42]. After the recovery of the TPW oscillation, East Asia returned to and remained at warm and humid midlatitudes (~40°N) during 140–130 Ma, where the favorable environment might have helped promote to the subsequent ca. 135 Ma flourishing of the even more prosperous Jehol Biota[50,51] (Fig. 4b).

The Late Jurassic–Early Cretaceous TPW oscillation coincides with the Jurassic–Cretaceous extinction[48] and therefore causal links with TPW may help explain this enigmatic extinction. While quantitative studies of extinction rate in the 1980s demonstrated that biotic turnover did occur, this boundary was never classed as a "major" (>50%) generic turnover event[52–54], but with 30% extinction of genera and 5% extinction of families[52] (Fig. 4e). It is also essentially an "extinction without a cause" as multiple authors have tried, unconvincingly, to invoke the two most common causes of "major" mass extinction, greenhouse gas increase due to flood basalt, or asteroid impact. Increasing the problem has been the difficulty of correlation because of the observed changes in faunal realms as well as the enigmatic, Berriasian Stage erosional vacuity and hiatus caused by rapid sea level drops. A more parsimonious explanation may be TPW, which can explain the palaeobiogeographic changes, where boreal and tropical faunas underwent rapid reorganization, as well as the sea level event, itself more rapid than most events of its magnitude.

TPW affects diversity directly by reorienting ecospace relative to the latitudinal diversity gradient, as well as indirectly through relative sea-level change[54–58]. Earth has a fundamentally zonal climate structure that varies by latitude, and plates move quickly during the TPW period across different zonal climate belts[25] can significantly affect the living ecosystems of different species globally. As much of the Jurassic-Cretaceous tetrapod signal (these species being keenly sensitive to sea-level change) is exclusively European, it is the Boreal realm that is most important to assess whether the predictions made by TPW are consistent with observed biodiversity patterns or not (Fig. 4). During the "monster shift" excursion, the Boreal realm shifted to lower latitudes (Fig. 4c), predicting origination and sea-level rise creating more ecospace in shallow water, which are both consistent with the observed Late Jurassic increase in biodiversity (Fig. 4e). Then, during the "monster shift" recovery, the Boreal realm shifted back to higher latitudes (Fig. 4d), predicting extinction and sea-level drop restricting shallow water ecospace, which are both consistent with the boundary extinction (Fig. 4e). Thus, the Jurassic-Cretaceous TPW oscillation revealed in full here can potentially explain not only the extinction across the boundary, but also the rise in biodiversity in the Late Jurassic before it.

## Methods
### Zircon U–Pb geochronology
U–Pb dating of zircon was conducted using both SIMS and LA-ICP-MS to provide age constraints for the palaeomagnetic sampling sections. Zircons were separated by conventional magnetic and heavy liquid methods before hand-picking under a binocular microscope. More than 200 zircon grains from each of the five samples (TCZ-TA1, TCZ-TB1, TCZ-M1, TCZ-M2 and TSJ-TC1) were mounted in epoxy resin, which was polished and coated to reveal zircon cores. All zircons were documented with transmitted and reflected light micrographs as well as cathodoluminescence (CL) images to reveal their internal structures, and the mount was vacuum-coated with high-purity gold prior to SIMS analyses.

For every volcanic or pyroclastic sample (TCZ-TA1, TCZ-TB1, TCZ-M2 and TSJ-TC1), about 20–40 zircon U–Pb analyses were conducted on the CAMECA IMS-1280HR SIMS at the Institute of Geology and Geophysics, Chinese Academy of Sciences (IGGCAS) in Beijing. The operating and data-processing procedures used were similar to those described by Li et al.[59]. The U–Pb concentrations and isotopic compositions were calibrated against zircon standard ZS and a second

standard, GBW04705 (Qinghu), was also analyzed with the zircon grains as an unknown[60]. A long-term uncertainty for the $^{206}Pb/^{238}U$ measurements of the standard zircons was propagated to the unknowns[61] (1 relative standard deviation (RSD) = 1.5%). Measured compositions were corrected for common Pb using non-radiogenic $^{204}Pb$ and an average of present-day crustal composition[62]. Uncertainties on individual analyses in data tables are reported at a 1σ level; mean ages for pooled U/Pb analyses are quoted with a 95% confidence interval. Age calculations were carried out using the Isoplot/Ex v. 4.1 program[63]. All results were listed in Supplementary Table 2.

Zircon U–Pb dating (TCZ-M1) using LA-ICP-MS was conducted on an Agilent 7700e ICP-MS instrument, employing a COMPexPro 102 ArF excimer laser and a Microlas optical system at the Wuhan SampleSolution Analytical Technology Co., Ltd., Wuhan, China. Zircon 91500 and glass NIST610 were used as external standards for U–Pb dating calibration. An Excel-based software ICPMSDataCal was used to perform off-line selection and integration of background and analyzed signals, time-drift correction and quantitative calibration for U–Pb dating[64]. Concordia diagrams and weighted mean calculations were made using ISOPLT4.1 software[63]. The results that younger than 350 Ma were listed in Supplementary Table 2.

### Rock magnetism

Samples exhibiting representative demagnetization behavior from each section were chosen for rock magnetic measurements, including hysteresis loops, isothermal remanent magnetization (IRM) acquisition curves, and back-field demagnetization curves, which were measured with a MicroMag 3900 Vibrating Sample Magnetometer (Princeton Measurements Corp., USA). Coercivity ($B_c$), remanent magnetization ($M_{rs}$), and saturation magnetization ($M_s$) were calculated after high-field slope correction. The coercivity of remanence ($B_{cr}$) was obtained by stepwise demagnetization of saturated isothermal remanent magnetization (SIRM) in a field up to 1 or 1.5 T. The unmixing IRM acquisition curves[65] were used to analyze the magnetic component composition. Anisotropy of magnetic susceptibility (AMS) was measured using an AGICO MFK-1FA Kappabridge before demagnetization.

### Palaeomagnetism

All palaeomagnetic specimens were subjected to stepwise thermal demagnetization using a PGL-100 thermal demagnetizer, in which the residual magnetic field is minimized to 1 nT (ref. 66). Thermal demagnetization was applied progressively in 16−20 steps, with temperature intervals of 10−80 °C. Remanent magnetizations were measured with a 2G-755 cryogenic magnetometer after each step of thermal demagnetization. Both the demagnetizer and the magnetometer are installed in a magnetically shielded room with a background field of <300 nT at the Palaeomagnetism and Geochronology Laboratory (PGL), Institute of Geology and Geophysics, Chinese Academy of Sciences (IGGCAS). Demagnetization results for each specimen were evaluated by principal component analysis[67] and the mean remanence directions were computed by Fisher statistics[68] using the PaleoMac software[69].

### Data availability

The palaeomagnetic data generated in this study have been deposited in the Open Science Framework database [https://osf.io/jnvwt/?view_only=d9fd8281982f410488e2b22f5d3539b1]. All data generated in this study are provided in the Supplementary Information file.

### Code availability

The PaleoMac software used for palaeomagentic analyses is available at https://www.ipgp.fr/~fluteau/. Elongation/inclination (E/I) correction were analyzed using the PmagPy (An open source package for palaeomagnetic data analysis) at https://pmagpy.github.io/PmagPy-docs/intro.html.

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

## Acknowledgements

We thank Youtang Liu, Jichang Zhu, Liang Dong, and Hu Tang for assistance in the field. We appreciated helpful discussions with Yongqing Liu at the Institute of Geology, Chinese Academy of Geological Sciences and Qingren Meng at the State Key Laboratory of Lithospheric Evolution, Institute of Geology and Geophysics, Chinese Academy of Sciences. This study is funded by the National Natural Science Foundation of China grants 42288201 (To R.-X.Z.), Strategic Priority Program (B) of the Chinese Academy of Sciences Grant No. XDB0710000 (to P.Z.), National Natural Science Foundation of China grants 92155203 (to P.Z.) and the National Science Fund for Young Scholars 42102015 (To W.-X.H.).

## Author contributions

P.Z., H.Q., and R. Z. designed research. Y.H., P.Z., W.H., M.Z., and J.Y. performed research. Y.H., P.Z., C.D., R.N.M. analyzed and interpreted the palaeomagnetic data. Q.L. contributed the SIMS zircon U–Pb dating. P.W. contributed the discussion about the environmental and palaeontological implications. The manuscript was drafted by Y.H and P.Z and edited by all authors.

## Competing interests

The authors declare no competing interests.
