## [Peer Review File · Nature Communications]

REVIEWER COMMENTS

Reviewer #1 (Remarks to the Author):

I think the geochronology and paleomagnetic work are both well-done and thorough. The Tuchengzi and Tiaojishan Formations are perfectly suited in emplacement age to address the paleomagnetic question and, while their intermediate-felsic and sedimentary lithologies may have complicated the paleomagnetic record, the combination of passing fold tests, rock magnetic characterization, and inclination shallowing correction are sufficient, in my opinion, to show that the data are more or less robust. I suggest a few major edits:

- In terms of data quality, the amount of paleomagnetic data with passing fold test makes for a fairly convincing dataset. The among-sample scatter is quite large, but the large N seems to result in reasonable mean directions. I'm a little wary of the procedure of rotating apparently anomalous sets of directions from the top of the Tiaojishan Fm into alignment based on the differences in paleomagnetic directions alone. This is a case of circular reasoning- one assumes that there was vertical axis rotation and so ascribes the discrepancy to vertical axis rotation in order to correct the vertical axis rotation. The authors should show some detailed geologic maps of the study area (as they do in Supplementary Fig. 2), but show explicitly where they think the vertical axis rotations are being accommodated. Without this kind of independent verification that there were such rotations, I would be quite uncomfortable with the resulting poles. Who is to say which of the directional groups are rotated and which ones faithful to the larger craton orientation? And what if all of the directions are rotated?
- This leads me to a second and related point- whatever the authors do with the geological context for vertical axis rotation, I very strongly recommend sensitivity analysis to explore the actual effect of the vertical axis rotations on the ultimate TPW motion constraints. Given the location of NCC almost 90° from the rotation axis, it might not move the ultimate answer much. However, the burden of proof is on the authors to show this. They should assume a range of plausible rotations and see how the TPW shifts (ie Fig. 4) change. At least some of this should be included in the main text due to its importance.
- A major issue is that there have been some important papers about the Jurassic "monster shift" that the MS does not address. Most importantly, a work by Vaes et al. published this year (A global apparent polar wander path for the last 320 Ma calculated from site-level paleomagnetic data). This is quite an extensive compilation with authors from many active paleomag groups, and they see no evidence for the Jurassic shift. I and the community would be very interested in how the authors respond to this. I can see a number of arguments in each direction, but will let the authors voice their thoughts in their paper.

Reviewer #2 (Remarks to the Author):

The Late Jurassic true polar wander (TPW, from ~160 Ma to ~145 Ma) has been demonstrated from both

continental and oceanic paleomagnetic data. But it has been controversial as to whether it involved rapid or steady rotations of lithosphere. Resolving the controversy requires high-resolution, high quality paleomagnetic data from the time interval in question.

The Late Jurassic TPW is manifested as a southward shift of the North China craton (NCC) from a higher latitude to a lower latitude, which is broadly supported by the existing paleomagnetic poles of the NCC at ~170 Ma, ~165 Ma, ~160 Ma, ~155 Ma, and ~140 Ma (Ren et al., 2018; Yi et al., 2019; Gao et al., 2022). To further constrain the duration and speed of the Late Jurassic TPW, Hou et al conducted a paleomagnetic study of the Late Jurassic to Early Cretaceous volcano- and clastic sequences in the NCC and obtained three new paleomagnetic poles at ~153 Ma, ~147 Ma, and ~141 Ma. These three new poles fill the data gap between the existing ~155 Ma and ~140 Ma poles, and thus can refine the TPW in terms of its duration and speed. The new paleomagnetic data show a ~12° southward shift in paleolatitude from ~153 Ma to ~147 Ma, followed by a ~10° northward shift from ~147 Ma to ~141 Ma. As such, the authors argue that the new data documents a complete round-trip TPW oscillation in Late Jurassic-Early Cretaceous. An oscillating TPW in the Late Jurassic would have had profound geodynamic and environmental consequences as the author argued. As such, this study represents an important contribution to the understanding of the Late Jurassic TPW.

My comments are mainly focused on the ~147 Ma paleomagnetic data for the following reasons.

1) The oscillating latitudinal shifts of the NCC suggested by the new paleomagnetic data are best illustrated in Fig. 4b. The ~153 Ma pole (F, Fig. 4b) yields a paleolatitude (~33°N) similar to those of the existing ~160 Ma and ~155 Ma poles (D, E, Fig. 4b). Similarly, the ~141 Ma pole yields a paleolatitude (~35°N) comparable to that of the existing ~140 Ma pole (I, Fig. 4b). The new ~147 Ma pole yields a much lower paleolatitude at ~25°N (G, Fig. 4b). It is this low latitude that results in the oscillating shifts of the NCC between ~153 Ma and ~141 Ma (Fig. 4b). Therefore, the argument for the complete round-trip TPW in this study rests heavily on the quality of the new ~147 Ma paleomagnetic data from the NCC.

2) The ~147 Ma paleomagnetic data are derived from the middle part of the Tuchengzi Formation (TCZ-M). Fig. 2a shows that the paleomagnetic samples were collected from both volcanic rocks and sandstones for the TCZ-M section. But only data from the red sandstones are mentioned in the text and shown in Table S4. Also, since a ChRM could potentially be contaminated by overprints of a chemical remanent magnetization (CRM) (Line 138), the rigor of the ~147 Ma ChRMs can be further demonstrated to ensure the ~147 Ma ChRMs are free from CRM contamination. Since no detailed demagnetization data of the TCZ-M samples are available, it is not possible to evaluate whether a defined ChRM is immune from CRM overprints or not. At the minimum, the temperature ranges over which a ChRM is defined need to be shown in Table S4. In addition, since the ChRMs of the TCZ-M samples show a wide range of inclinations ranging from ~10° to ~70° (Fig. 3b), more orthogonal vector plots the TCZ-M samples need to be shown in Fig. S5. In the current Fig. S5, the orthogonal vector plots of only two samples (TCZ-M24 and TCZ-M47) with apparently steep inclinations (>45°) of ChRMs are shown (Fig. S5d).

3) There appears inconsistency between the data shown in Fig. S5 and those summarized in Table S4 for the same samples. For instance, the orthogonal vector plots of the TCZ-M24 and TCZ-M47 samples appear to show steeper inclinations (>45°) of ChRMs (Fig. S5d), while in the data table the two samples

carry inclinations of 20.8° and 28.6°, respectively (Table S4). Also, looking at Fig. S5e, I notice that the orthogonal vector plot of the TCZ-TA51 sample shows a downward-pointing ChRM, but Table S4 shows that this sample has a negative inclination value of -43°, indicating an upward-pointing ChRM. Additionally, the TCZ-TA16 sample displays a well-defined ChRM in Fig. S5e, but the ChRM of this sample is not included in Table S4. Does this mean that the ChRM of the TCZ-TA16 sample was not included in the calculation of the mean for the TCZ-TA section? Please double check these data.

References:

Ren, Q., Zhang, S., Wu, Y., Yang, T., Gao, Y., Turbold, S., et al., 2018. New late Jurassic to early Cretaceous paleomagnetic results from North China and southern Mongolia and their implications for the evolution of the Mongol-Okhotsk suture. *Journal of Geophysical Research: Solid Earth*, 123, 10,370–10,398.

<https://doi.org/10.1029/2018JB016703>

Yi, Z., Liu, Y., and Meert, J.G., 2019. A true polar wander trigger for the Great Jurassic East Asian Aridification: *Geology*, v. 47, p. 1112–1116, <https://doi.org/10.1130/G46641.1>

Gao et al., 2021. North China block underwent simultaneous true polar wander and tectonic convergence in late Jurassic: New paleomagnetic constraints. *Earth and Planetary Science Letters* 567, 117012. <https://doi.org/10.1016/j.epsl.2021.117012>

Reviewer #3 (Remarks to the Author):

This paper provides new radiometric and paleomagnetic data from Jurassic rocks from China used to test a recently proposed hypothesis that the solid Earth underwent an episode of true polar wander (TPW) in the late Jurassic/early Cretaceous. The data presented appear to be of very good quality and, used in conjunction with data from the literature, allowed the construction of a detailed APWP for North China that confirms the existence of the Jurassic TPW 'monster shift' event. I have very little comments to make because this paper looks very solid and, very importantly, it is well organized, focused, and positively straightforward to read and comprehend: a testable hypothesis has been tested, and confirmed. Science should always be like this.

Anyway, I have just a few comments on the end chapters. In 'Mechanism of...(page 11, LINES 267-270), I don't follow: what are exactly the relations between a slab passing the MTZ and the equatorward vs. poleward TPW? how is ref.44 pertinent to this discussion? My understanding is that there could be piling up of slabs at the MTZ followed by rapid (avalanche) descent into the lower mantle, causing TPW. I do not completely get the 'reverse slab' mechanism to explain the TPW rebound. Perhaps this chapter should be made more clear. See also Li et al. (ESR, 2019).

In chapter 'Environmental and...', I am not sure about the 'quadrature' argument and the transgressions vs regressions issues. Can we just simply say that Earth has got a fundamentally zonal climate structure and that plates move across zonal climate belts (continental drift + TPW) such that if, say, you are initially at mid latitudes with lots of happy life you may end up as consequence of TPW in the middle of the arid belt where life is not happy and had to migrate to not die off entirely? sorry for the

inconspicuous wording but I don't get the TPW breaking the world into quadrants, I just see plain old plate stratigraphy in action!

In any case, congratulations, this is a good paper worth publishing.

Our responses to each point raised by reviewers are in **Blue** and actions for how we revised the manuscript are in **Red**. All lines numbers refer to the revised manuscript *without* tracked changes.

Reviewer #1 (Remarks to the Author):

I think the geochronology and paleomagnetic work are both well-done and thorough. The Tuchengzi and Tiaojishan Formations are perfectly suited in emplacement age to address the paleomagnetic question and, while their intermediate-felsic and sedimentary lithologies may have complicated the paleomagnetic record, the combination of passing fold tests, rock magnetic characterization, and inclination shallowing correction are sufficient, in my opinion, to show that the data are more or less robust. I suggest a few major edits:

We appreciate the informed reviewer taking time to carefully read and review this manuscript. We also are very glad to see the positive comments about our manuscript.

- In terms of data quality, the amount of paleomagnetic data with passing fold test makes for a fairly convincing dataset. The among-sample scatter is quite large, but the large N seems to result in reasonable mean directions. I'm a little wary of the procedure of rotating apparently anomalous sets of directions from the top of the Tiaojishan Fm into alignment based on the differences in paleomagnetic directions alone. This is a case of circular reasoning- one assumes that there was vertical axis rotation and so ascribes the discrepancy to vertical axis rotation in order to correct the vertical axis rotation. The authors should show some detailed geologic maps of the study area (as they do in Supplementary Fig. 2), but show explicitly where they think the vertical axis rotations are being accommodated. Without this kind of independent verification that there were such rotations, I would be quite uncomfortable with the resulting poles. Who is to say which of the directional groups are rotated and which ones faithful to the larger craton orientation? And what if all of the directions are rotated?

We understand the reviewer's concern that the vertical-axis rotation should not be based on the differences in palaeomagnetic directions alone. For better explanation, we compiled the declination data of all reported palaeopoles of the NCC between the 130 Ma and 180 Ma (see the figure below; Data from Supplementary Table 5). One can see that the declinations of TJS-TA and the 155 Ma palaeopole (from Yi et al., 2019) are different others, whereas the declinations of TJS-TB and TJS-TC are in the same range of others. The stable declinations of 180-130 Ma poles indicate that the palaeo-position of the NCC as a whole is relatively stable in this time period without obvious self-rotation. Therefore, it is reasonable to ascribe the big declination difference of TJS-TA and 155 Ma palaeopole (E in the figure) to local vertical-axis rotation caused by local

faults. In the supplementary figure 2, we show these local faults, most of which are thrust and strike-slip faults due to Yanshan orogenic movement. Our sampling locations are in the northern margin of the NCC, where was strongly affected by the Yanshan orogenic movement with many regional and local faults such as the Chengde and Chengde Country faults, that can produce some anomalous declinations.

To further bolster our arguments about vertical-axis rotation, we added two sentences in Lines 124-129 to address this problem. And added the following figure showing different declinations as a supplementary figure 7.

- This leads me to a second and related point- whatever the authors do with the geological context for vertical axis rotation, I very strongly recommend sensitivity analysis to explore the actual effect of the vertical axis rotations on the ultimate TPW motion constraints. Given the location of NCC almost 90° from the rotation axis, it might not move the ultimate answer much. However, the burden of proof is on the authors to show this. They should assume a range of plausible rotations and see how the TPW shifts (ie Fig. 4) change. At least some of this should be included in the main text due to its importance.

Maybe our discussion of rotation was misleading. We use local vertical-axis rotation to explain the declination difference between our section TJS-TA and other two sections of the Tiaojiashan Formation in this study. This kind of local vertical-axis rotation is mainly caused by local strike-slip faulting, which can affect declination of the palaeomagnetic direction but will not affect the palaeolatitude of the NCC as vertical-axis rotation will not influence inclination. (A similar argument was made for using data from the vertically-rotated La Negra volcanics of South America as they still yielded meaningful paleolatitudes to help test the “monster shift”; Fu et al., 2020). Therefore, vertical-axis rotation affecting our declination data does not affect the palaeolatitude of the NCC that is used to infer the TPW motion. As the location of NCC is almost 90° from the rotation axis, the TPW rotation would

have resulted in a large latitudinal movement of the NCC in a short time period. As mentioned, vertical-axis rotation only affects our declination data and will not affect the palaeolatitude of the NCC, which means it will not affect our discussion about the ultimate TPW motion constraints.

We added a sentence to address this problem the revised manuscript (lines 132-133).

- A major issue is that there have been some important papers about the Jurassic “monster shift” that the MS does not address. Most importantly, a work by Vaes et al. published this year (A global apparent polar wander path for the last 320 Ma calculated from site-level paleomagnetic data). This is quite an extensive compilation with authors from many active paleomag groups, and they see no evidence for the Jurassic shift. I and the community would be very interested in how the authors respond to this. I can see a number of arguments in each direction but will let the authors voice their thoughts in their paper.

We appreciate that the reviewer recommends to us the new work. In the original version, we discussed several papers that didn’t recognized this “monster shift” TPW event (e.g., Torsvik et al., 2012 ESR, 2014; PNAS; Kulakov et al., 2021). The new work of Vaes et al., 2023 provide a new global APWP for the last 320 million years that is calculated from simulated site-level palaeomagnetic data instead of from palaeopoles. The new method could incorporate spatial and temporal uncertainties of the original datasets and create APWP with better smoothing and less error. Meanwhile, the new APWP is established in a 20 Myr window. This long-time window does not have the resolution for recognizing a rapid TPW event such as is being investigated here. Such a putatively rapid TPW excursion is a transient event, which can be easily smoothed out by such a protracted time window. The TPW “monster shift” is thought to have occurred between 160 and 140 Ma, which means the whole duration of this TPW event is within the time-window of the new APWP. The 20 Myr- resolution-APWP is far below the resolution needed to recognize the TPW “monster shift” in the Late Jurassic–Early Cretaceous. Only successive palaeopoles within the 160–140 Ma window can achieve this and this is precisely what we do in this study.

We added two sentences in Lines 236-239 to address this APWP and why it cannot recognize the TPW “monster shift” as follows.

“Furthermore, a new global APWP calculated from site-level data also did not recognize the “monster shift” either (Vaes et al., 2023). However, as this new APWP was calculated in a 20 Myr window, which is even longer than the whole duration of the “monster shift” TPW, the new APWP is below the resolution needed to reveal the rapid TPW event.”

Reviewer #2 (Remarks to the Author):

The Late Jurassic true polar wander (TPW, from ~160 Ma to ~145 Ma) has been demonstrated from both continental and oceanic paleomagnetic data. But it has been controversial as to whether it involved rapid or steady rotations of lithosphere. Resolving the controversy requires high-resolution, high quality paleomagnetic data from the time interval in question.

The Late Jurassic TPW is manifested as a southward shift of the North China craton (NCC) from a higher latitude to a lower latitude, which is broadly supported by the existing paleomagnetic poles of the NCC at ~170 Ma, ~165 Ma, ~160 Ma, ~155 Ma, and ~140 Ma (Ren et al., 2018; Yi et al., 2019; Gao et al., 2022). To further constrain the duration and speed of the Late Jurassic TPW, Hou et al. conducted a paleomagnetic study of the Late Jurassic to Early Cretaceous volcano- and clastic sequences in the NCC and obtained three new paleomagnetic poles at ~153 Ma, ~147 Ma, and ~141 Ma. These three new poles fill the data gap between the existing ~155 Ma and ~140 Ma poles, and thus can refine the TPW in terms of its duration and speed. The new paleomagnetic data show a ~12° southward shift in paleolatitude from ~153 Ma to ~147 Ma, followed by a ~10° northward shift from ~147 Ma to ~141 Ma. As such, the authors argue that the new data documents a complete round-trip TPW oscillation in Late Jurassic–Early Cretaceous. An oscillating TPW in the Late Jurassic would have had profound geodynamic and environmental consequences as the author argued. As such, this study represents an important contribution to the understanding of the Late Jurassic TPW.

We appreciate the reviewer taking time to carefully read and review this manuscript. We are also very glad to read the positive comments on our manuscript.

My comments are mainly focused on the ~147 Ma paleomagnetic data for the following reasons.

1) The oscillating latitudinal shifts of the NCC suggested by the new paleomagnetic data are best illustrated in Fig. 4b. The ~153 Ma pole (F, Fig. 4b) yields a paleolatitude (~33°N) similar to those of the existing ~160 Ma and ~155 Ma poles (D, E, Fig. 4b). Similarly, the ~141 Ma pole yields a paleolatitude (~35°N) comparable to that of the existing ~140 Ma pole (I, Fig. 4b). The new ~147 Ma pole yields a much lower paleolatitude at ~25°N (G, Fig. 4b). It is this low latitude that results in the oscillating shifts of the NCC between ~153 Ma and ~141 Ma (Fig. 4b). Therefore, the argument for the complete round-trip TPW in this study rests heavily on the quality of the new ~147 Ma paleomagnetic data from the NCC.

Yes, as the reviewer mentioned, the ~147 Ma paleomagnetic pole is one of the most important poles in our work. Based on this pole, we proposed a complete round-trip TPW which represents a further step for studies of the Late Jurassic–Early Cretaceous TPW.

2) The ~147 Ma paleomagnetic data are derived from the middle part of the Tuchengzi Formation (TCZ-M). Fig. 2a shows that the paleomagnetic samples were collected from both volcanic rocks and sandstones for the TCZ-M section. But only data from the red sandstones are mentioned in the text and shown in Table S4. Also, since a ChRM could potentially be contaminated by overprints of a chemical remanent magnetization (CRM) (Line 138), the rigor of the ~147 Ma ChRMs can be further demonstrated to ensure the ~147 Ma ChRMs are free from CRM contamination. Since no detailed demagnetization data of the TCZ-M samples are available, it is not possible to evaluate whether a defined ChRM is immune from CRM overprints or not. At the minimum, the temperature ranges over which a ChRM is defined need to be shown in Table S4. In addition, since the ChRMs of the TCZ-M samples show a wide range of inclinations ranging from ~10° to ~70° (Fig. 3b), more orthogonal vector plots the TCZ-M samples need to be shown in Fig. S5. In the current Fig. S5, the orthogonal vector plots of only two samples (TCZ-M24 and TCZ-M47) with apparently steep inclinations (>45°) of ChRMs are shown (Fig. S5d).

We appreciate the reviewer taking time to carefully read and provide some very useful questions. The Tuchengzi Formation is mainly composed of red sandstone and conglomerate and we are very lucky to identify a layer of volcanic rocks (about 3-meter thick) within the red sandstone layers. We sampled ten cores from this pyroclastic rock layer, however the ten cores displayed dispersed directions that were inconsistent with directions from the red sandstone layers (see following figure). The dispersed directions may be influenced by breccia within the pyroclastic rocks, therefore, they were excluded from further calculations. For comparison, we also add the data of the ten volcanic samples in Supplementary Table 4.

About the ~147 Ma pole, we argue that the CRM only affected the LTC of the TCZ-M samples, whereas the ChRMs (HTC) were not affected by CRM overprints for the following reasons. (i) The temperature ranges for the ChRMs are defined higher than 400 °C, and higher than 600 °C for most samples (Supplementary Table 4). (i) The ChRMs of the TCZ-M samples display antipodal normal and reversal polarities; if a CRM affected the ChRMs, there will only one polarity. So, we argue that our ChRMs are unaffected by Chemical overprinting.

We follow the reviewer's suggestion and added another 4 orthogonal vector plots of the TCZ-M samples in Supplementary Fig. 5 to display our results.

We added temperature ranges of ChRMs from TCZ-M section in the Supplementary Table 4 and another 4 orthogonal vector plots of the TCZ-M samples in Supplementary Figure 5. For comparison, we also add the data of the ten volcanic rocks in Supplementary Table 4. We also added two sentence in Lines 153-155 to address this problem.

3) There appears inconsistency between the data shown in Fig. S5 and those summarized in Table S4 for the same samples. For instance, the orthogonal vector plots of the TCZ-M24 and TCZ-M47 samples appear to show steeper inclinations ($>45^\circ$) of ChRMs (Fig. S5d), while in the data table the two samples carry inclinations of 20.8° and 28.6° , respectively (Table S4). Also, looking at Fig. S5e, I notice that the orthogonal vector plot of the TCZ-TA51 sample shows a downward-pointing ChRM, but Table S4 shows that this sample has a negative inclination value of -43° , indicating an upward-pointing ChRM. Additionally, the TCZ-TA16 sample displays a well-defined ChRM in Fig. S5e, but the ChRM of this sample is not included in Table S4. Does this mean that the ChRM of the TCZ-TA16 sample was not included in the calculation of the mean for the TCZ-TA section? Please double check these data.

We appreciate your careful reading of the manuscript and pointing out our mistakes. We made this mistake when preparing Supplementary Figure 5. We want to show the best orthogonal vector plot and changed this figure several times, but forgot to change the sample names. The name of "TCZ-M24 and TCZ-M47" samples should be TCZ-M14 and TCZ-M29. In the new version, we changed these samples names. For the "TCZ-TA51" sample, we also marked the wrong name. Its name should be TCZ-TA46. The picture following show orthogonal vector plot of the TCZ-TA51, whose inclination value is negative. We also include this plot of the TCZ-TA51 in Supplementary Figure 5. The label "TCZ-TA16" should be corrected to TCZ-TA11. We have corrected the mistake in the revised manuscript and double-checked all data. Thanks for catching this!

We double-checked all data and corrected the wrong sample names to the right ones in Supplementary Figure 5. We also include this additional sample (TCZ-TA51) in Supplementary Figure 5.

Reviewer #3 (Remarks to the Author):

This paper provides new radiometric and paleomagnetic data from Jurassic rocks from China used to test a recently proposed hypothesis that the solid Earth underwent an episode of true polar wander (TPW) in the late Jurassic/early Cretaceous. The data presented appear to be of very good quality and, used in conjunction with data from the literature, allowed the construction of a detailed APWP for North China that confirms the existence of the Jurassic TPW 'monster shift' event. I have very little comments to make because this paper looks very solid and, very importantly, it is well organized, focused, and positively straightforward to read and comprehend: a testable hypothesis has been tested, and confirmed. Science should always be like this.

We appreciate the reviewer taking time to carefully read and review this manuscript. And we are very happy to have your positive comments.

Anyway, I have just a few comments on the end chapters. In 'Mechanism of...(page 11, LINES 267-270), I don't follow: what are exactly the relations between a slab passing the MTZ and the equatorward vs. poleward TPW? how is ref.44 pertinent to this discussion? My understanding is that there could be piling up of slabs at the MTZ followed by rapid (avalanche) descent into the lower mantle, causing TPW. I do not

completely get the 'reverse slab' mechanism to explain the TPW rebound. Perhaps this chapter should be made more clear. See also Li et al. (ESR, 2019).

We made a mistake with the reference 44. This should be another paper of Steinberger et al., 2017, which is related to this discussion. Steinberger et al., 2017 modeled how subducted slabs drive TPW. Because the degree-2 geoid kernel (the following figure) is positive in the upper part of the mantle, the effect of upper mantle slabs is to be shifted toward the equator. In contrast, the kernel (the following Figure from Steinberger et al., 2017) is negative in the lower part of the mantle. Therefore, for slabs in the lower part of the mantle, the effect is to be shifted toward to pole. Therefore, the location of subducted slab will affect the inertia tensor and cause equatorward and poleward TPW. Here, we use one sentence to show this kind of phenomena and then we explained how subducted slabs affect the round-trip TPW we studied.

[Redacted]

In the new version, we changed reference 44 to Steinberger et al., 2017.

In chapter 'Environmental and...', I am not sure about the 'quadrature' argument and the transgressions vs regressions issues. Can we just simply say that Earth has got a fundamentally zonal climate structure and that plates move across zonal climate belts (continental drift + TPW) such that if, say, you are initially at mid latitudes with lots of happy life you may end up as consequence of TPW in the middle of the arid belt where life is not happy and had to migrate to not die off entirely? sorry for the inconspicuous wording but I don't get the TPW breaking the world into quadrants, I just see plain old plate stratigraphy in action!

Thanks for your valuable questions. Following you suggestion, we use the simply way that just say that Earth has a fundamentally zonal climate structure and that plates move across zonal climate belts. We have changed the last paragraph accordingly and delete the discussion about "quadrature".

In the new version, we use a simple approach, just to say that Earth has a fundamentally zonal climate structure and that plates move across zonal climate belts. We have changed the last paragraph and delete the discussion about "quadrature". We also delete the "quadrature" figure in figure 4 and add fundamental climate zones in Figure 4 instead.

In any case, congratulations, this is a good paper worth publishing.

Thanks again for your positive comments about this manuscript.

REVIEWER COMMENTS

Reviewer #1 (Remarks to the Author):

Dear Authors,

Thank you for your thoughtful reply to my concerns about your manuscript. I understand that the vertical axis rotation is not likely to strongly affect the ultimate implications for TPW, since the latter is mainly an inclination signal. I would still suggest a quantitative effort on this above and beyond the extra sentence "It is worth noting that this vertical-axis rotation does not affect the palaeolatitude of the NCC, and therefore, does not influence the discussion of TPW."

It should not take much effort to compute, given the proposed TPW longitudes for example in Torsvik and/or Fu and Kent's works, what are the predicted changes in declination (unless your site is EXACTLY 90° away in longitude, the answer is not "zero") and how does that compare to your uncertainties in declination, both nominal and including the contribution from unquantified vertical axis rotations? It seems like you are not using the full information available in your dataset otherwise!

Also, note that the Vaes et al study uses 20 My windows, but does them at 10 My intervals. Is it really true that this ~15 My event would not be picked up? If there is a shift over 15 My, I would think the two 10 My-spaced means just before and after the center of the event would see some shift, even in 20 My windows since nearly 50% of each 20 My window would include Monster Shifted poles. So I find the binning resolution a little hard to believe as the only explanation. I strongly recommend that the authors more quantitatively demonstrate that this is the issue by, for example, running 10 My-spaced, 20 My windowed binning on their data and see what they get. It would be in the authors' interest to be careful with this argument as the Vaes et al. compilation has, in my view, gotten a lot of traction in the community and, if the authors do not address this in more detail now, this will be raised as a potential counterargument to their result in the near future.

Reviewer #2 (Remarks to the Author):

My previous comments were focused on the data from the TCZ-M. In the revision, the authors have added the paleomagnetic data from the pyroclastic rock layer of TCZ-M in Table S4, provided demagnetization temperature ranges for ChRMs from TCZ-M in the data table, and corrected the previously mistaken labels of the orthogonal plots in Fig. S5. In my view, the manuscript is now in a good shape.

Our responses to each point raised by reviewers are in **Blue** and our actions for how we revised the manuscript are in **Red**. All lines numbers refer to the revised manuscript *with* tracked changes.

Reviewer #1:

Dear Authors,

Thank you for your thoughtful reply to my concerns about your manuscript. I understand that the vertical axis rotation is not likely to strongly affect the ultimate implications for TPW, since the latter is mainly an inclination signal. I would still suggest a quantitative effort on this above and beyond the extra sentence "It is worth noting that this vertical-axis rotation does not affect the palaeolatitude of the NCC, and therefore, does not influence the discussion of TPW."

It should not take much effort to compute, given the proposed TPW longitudes for example in Torsvik and/or Fu and Kent's works, what are the predicted changes in declination (unless your site is EXACTLY 90° away in longitude, the answer is not "zero") and how does that compare to your uncertainties in declination, both nominal and including the contribution from unquantified vertical axis rotations? It seems like you are not using the full information available in your dataset otherwise!

Response: Thank you very much for reviewing our manuscript again and for this useful suggestion. We agree that an effort to quantitatively address the possibility of vertical axis rotation and TPW would be better for the discussion. Following your suggestion, we calculated the predicted changes in declination according to the global APWPs of both Kent and Torsvik (in the North China craton [NCC] coordinates with reference point at 41°E, 121°E) and compared it with our declination data from cross-section TJS-TA and cross-sections TJS-TB and TJS-TC (see the following figure). The global APWP computed by both Kent and Torsvik display clockwise rotation of the declination during the Late Jurassic TPW (160–145 Ma), similar to data from the NCC (see the following figure). The predicted declination change using APWP computed by Kent is $17^\circ \pm 4^\circ$ from 160–156 Ma caused by the Late Jurassic TPW. On the contrary, the APWP of Torsvik gives a predicted declination change of $6^\circ \pm 4^\circ$ from 160–150 Ma. The NCC was far away from the TPW Euler pole of rotation, which shows mainly latitudinal movement and a declination change of only $9^\circ \pm 4^\circ$ from 160–153 Ma (when we use the mean declination calculated from the TJS-TB and TJS-TC sections; shown in the figure below). This mean declination of TJS-TB and TJS-TC is within the range of the predicted declination change of Torsvik (no TPW) and Kent (with TPW), and consistent with the trend of declination changes. In contrast, the declination calculated from section TJS-TA indicates a large declination change of $35^\circ \pm 3^\circ$, which is much larger than the predicted declination change of Kent with TPW. Therefore, TPW cannot explain the large declination change of section TJS-TA. This observation thus provides an additional line of reasoning for why we ascribe this odd declination change of the TJS-TA data to local vertical-axis rotation relative to TJS-TB and TJS-TC.

Revision: For a better explanation of the various changes in declination observed, we changed Supplementary Figure 7 with the following figure. To make the statement more accurate, we modified sentences in Lines 132-139 as following:

“Comparing with the 160 Ma poles of the NCC, the mean declination calculated from sections TJS-TB and TJS-TC show a declination change of $9^\circ \pm 4^\circ$, which is consistent with predicted declination changes with APWPs from both Kent et al.⁸ ($17^\circ \pm 4^\circ$) and Torsvik et al.¹⁶ ($6^\circ \pm 4^\circ$; Supplementary Fig. 7). On the contrary, the data from section TJS-TA show a much larger declination change ($35^\circ \pm 3^\circ$), which is inconsistent with the declination change in the TPW framework (Supplementary Fig. 7). Therefore, we ascribe the declination change of section TJS-TA to local vertical axis rotation.”

Also, note that the Vaes et al study uses 20 My windows, but does them at 10 My intervals. Is it really true that this ~15 My event would not be picked up? If there is a shift over 15 My, I would think the two 10 My-spaced means just before and after the center of the event would see some shift, even in 20 My windows since nearly 50% of each 20 My window would include Monster Shifted poles. So I find the binning resolution a little hard to believe as the only explanation. I strongly recommend that the authors more quantitatively demonstrate that this is the issue by, for example, running 10 My-spaced, 20 My windowed binning on their data and see what they get. It would be in the authors' interest to be careful with this argument as the Vaes et al. compilation has, in my view, gotten a lot of traction in the community and, if the authors do not address this in more detail now, this will be raised as a potential counterargument to their result in the near future.

Response: We appreciate your careful reading and helpful advice. You give an

informed and good suggestion about addressing the recent APWP of Vaes et al. (2023).

1) In their paper, Vaes et al. compared their APWP results in both 10 Myr and 20 Myr averaging windows (their Figure 9). In Figure 9b, one can see the APW rate at ca. 150 Ma is higher for the 10 Myr window ($\sim 0.6^\circ/\text{Myr}$) than that of the 20 Myr window ($\sim 0.2^\circ/\text{Myr}$)—i.e., with rate being window-dependent, they may have smoothed out the sudden motion. The authors themselves admit that their APWP does not rule out the occurrence of TPW.

2) The Late Jurassic “monster shift” proposed by Kent et al. (2015) is mainly supported by several paleomagnetic poles: the 169 Ma Moat pole that overlaps the 190–160 Ma “standstill” poles; the 147 Ma Ithaca pole that is located over $\sim 30^\circ$ away from the Moat pole; and the 155 Ma Peddie pole that lies about midway between the Moat and Ithaca poles. All these three poles are from igneous rocks, however, they were not included in Vaes et al. (2023)’s computation for the new APWP. Without these three key poles, the new APWP cannot show the “monster shift” TPW. This data selection (omission) will cause some bias about the Late Jurassic TPW.

Revision: In the revised version, we add two sentences to address these two points in Lines 243-251 as following:

“However, in this new study, the APWP calculated in a 10 Myr window shows a higher APW rate ($\sim 0.6^\circ/\text{Myr}$) than that in a 20 Myr window ($\sim 0.2^\circ/\text{Myr}$) around 150 Ma, and the authors admitted that the new APWP does not rule out the occurrence of TPW in the Late Jurassic. Meanwhile, their APWP did not include three key poles (the 169 Ma Moat pole, the 155 Ma Peddie pole and the 147 Ma Ithaca pole) that support the “monster shift” TPW. Therefore, their not identifying Late Jurassic TPW can be ascribed to the data selection (and omission) and a time window used that was below the resolution needed to reveal the “monster shift” TPW.”.

Reviewer #2 (Remarks to the Author):

My previous comments were focused on the data from the TCZ-M. In the revision, the authors have added the paleomagnetic data from the pyroclastic rock layer of TCZ-M in Table S4, provided demagnetization temperature ranges for ChRMs from TCZ-M in the data table, and corrected the previously mistaken labels of the orthogonal plots in Fig. S5. In my view, the manuscript is now in a good shape.

Response: Thank you for your detailed read and positive assessment of the manuscript.

REVIEWERS' COMMENTS

Reviewer #1 (Remarks to the Author):

Hi,

I think the revised manuscript is much better in terms of quantifying the declination rotation. However, this sentence is obviously problematic:

"the mean declination calculated from sections TJS-TB and TJS-TC show a declination change of $9^\circ \pm 4^\circ$, which is consistent with predicted declination changes with APWPs from both Kent et al.8 ($17^\circ \pm 4^\circ$) and Torsvik et al.16 ($6^\circ \pm 4^\circ$; Supplementary Fig. 7)."

9 ± 4 is obviously not "consistent" with 17 ± 4 . I wonder if there's a typo somewhere? If these are 1 sigma uncertainties instead of the standard 2 sigma used in paleomagnetism (and most of the geosciences) the authors should state that (or just use 2 sigma.). If these are already 2 sigma uncertainties, then the statement that these two values are consistent is false and needs to be revised. Maybe the authors can call on another local rotation.

Our responses to each point raised by reviewers are in **Blue** and our actions for how we revised the manuscript are in **Red**. All line numbers refer to the revised manuscript *with* tracked changes.

Reviewer #1 (Remarks to the Author):

Hi,

I think the revised manuscript is much better in terms of quantifying the declination rotation. However, this sentence is obviously problematic:

"the mean declination calculated from sections TJS-TB and TJS-TC show a declination change of $9^\circ \pm 4^\circ$, which is consistent with predicted declination changes with APWPs from both Kent at al.⁸ ($17^\circ \pm 4^\circ$) and Torsvik et al.¹⁶ ($6^\circ \pm 4^\circ$; Supplementary Fig. 7)."

9 ± 4 is obviously not "consistent" with 17 ± 4 . I wonder if there's a typo somewhere? If these are 1 sigma uncertainties instead of the standard 2 sigma used in paleomagnetism (and most of the geosciences) the authors should state that (or just use 2 sigma.). If these are already 2 sigma uncertainties, then the statement that these two values are consistent is false and needs to be revised. Maybe the authors can call on another local rotation.

Response: We appreciate your careful reading again and pointing out the wrong expression. As the reviewer said, "consistent" is not suitable here, we have modified relative expressions in the new manuscript. The palaeolatitudes calculated are already 2 sigma as standard.

Revision: In the revised version, we modified this sentence in Line 143-145 as following: "Comparing with the 160 Ma poles of the NCC, the mean declination calculated from sections TJS-TB and TJS-TC show a declination change of $9^\circ \pm 4^\circ$, which is in the same range as predicted declination changes with APWPs from both Kent at al.⁸ ($17^\circ \pm 4^\circ$) and Torsvik et al.¹⁶ ($6^\circ \pm 4^\circ$) when errors are considered (Supplementary Fig. 7).